# Intercropping on Mars: A promising system to optimise fresh food production in future martian colonies

Rebeca Gonçalves[1]*, G. W. Wieger Wamelink[2], Peter van der Putten[1], Jochem B. Evers[1]

1 Centre for Crop System Analysis, Wageningen University & Research, Wageningen, The Netherlands,
2 Wageningen Environmental Research, Wageningen University & Research, Wageningen, The Netherlands

* rebeca.rgoncalves@gmail.com

**Data Availability Statement:** All relevant data are within the manuscript.

**Funding:** The author(s) received no specific funding for this work.

## Abstract

Future colonists on Mars will need to produce fresh food locally to acquire key nutrients lost in food dehydration, the primary technique for sending food to space. In this study we aimed to test the viability and prospect of applying an intercropping system as a method for soil-based food production in Martian colonies. This novel approach to Martian agriculture adds valuable insight into how we can optimise resource use and enhance colony self-sustainability, since Martian colonies will operate under very limited space, energy, and Earth supplies. A likely early Martian agricultural setting was simulated using small pots, a controlled greenhouse environment, and species compliant with space mission requirements. Pea (*Pisum sativum*), carrot (*Daucus carota*) and tomato (*Solanum lycopersicum*) were grown in three soil types ("MMS-1" Mars regolith simulant, potting soil and sand), planted either mixed (intercropping) or separate (monocropping). Rhizobia bacteria (*Rhizobium leguminosarum*) were added as the pea symbiont for Nitrogen-fixing. Plant performance was measured as above-ground biomass (g), yield (g), harvest index (%), and Nitrogen/Phosphorus/Potassium content in yield (g/kg). The overall intercropping system performance was calculated as total relative yield (RYT). Intercropping had clear effects on plant performance in Mars regolith, being beneficial for tomato but mostly detrimental for pea and carrot, ultimately giving an overall yield disadvantage compared to monocropping (RYT = 0.93). This effect likely resulted from the observed absence of Rhizobia nodulation in Mars regolith, negating Nitrogen-fixation and preventing intercropped plants from leveraging their complementarity. Adverse regolith conditions—high pH, elevated compactness and nutrient deficiencies—presumably restricted Rhizobia survival/nodulation. In sand, where more favourable soil conditions promoted effective nodulation, intercropping significantly outperformed monocropping (RYT = 1.32). Given this, we suggest that with simple regolith improvements, enhancing conditions for nodulation, intercropping shows promise as a method for optimising food production in Martian colonies. Specific regolith ameliorations are proposed for future research.

**Competing interests:** The authors have declared that no competing interests exist.

## Introduction

As we enter a new age of space exploration, having a permanent settlement on Mars is now a reality in the not-so-distant future. Conquering the red planet is an exciting endeavour, but also one that brings several scientific, economic, and strategic benefits to our own planet and species.

Whilst rovers have begun to pave our way on Mars, the next stage of exploration and research would need to involve sending humans to perform the work themselves, as humans are much more autonomous and efficient at accomplishing field tasks and carrying out research [1]. However, having humans on Mars means they need to have their basic necessities covered, and this includes food. To supply and resupply from Earth all the food needs of a long-term settlement is both impractical as well as economically inviable, which means that the crew will have to make use of local Martian resources to produce at least part of their food needs locally [2].

The production of fresh food would in fact be an indispensable factor for any Martian colony. A bioregenerative food system, one where all resources are provided and produced *in situ* with minimal or no resupply from Earth, carries many advantages over the pre-packaged dehydrated foods that astronauts usually eat. As well as having the need for less food to be brought aboard the spacecraft (a factor that also significantly reduces the financial cost of the mission), a chief advantage of fresh food over dehydrated food is the retainment of nutrients essential to human health, especially antioxidants such as vitamin C and β-carotene, both of which are partially reduced or completely destroyed in the process of dehydration [3–5]. Although supplementary pills could be considered, fresh food provides a much better bioavailability of nutrients in the body, mediated via a complex array of other compounds present [6], and it further decreases the need to rely on resupplies from Earth for essential dietary requirements. The act of growing fresh food itself can also have a great positive impact on the mental well-being of colonists. Research on long-duration space missions, as well as analogue isolation simulations on Earth, have identified psychological and psychosocial factors as two of the most critical problems astronauts will face in their off-world missions [7,8]. Practising gardening activities has been repeatedly shown to be a significant and effective reliever of stress, anxiety and depression [9,10], and astronauts themselves have in the past expressed deep satisfaction at tending to plants in space [11].

### Prospects and challenges of Martian agriculture

Several methods for implementing bioregenerative food systems on Mars have been studied, most of which were focused on artificial growing media such as hydroponics and aquaponics. However, soil-based systems, which would involve making use of the available *in situ* Martian regolith, have recently come to new light as having fundamental features for the long-term sustainability of a colony on Mars. Such a system confers many practical and economic advantages over artificial growing media, such as offering better buffering capacity for operational errors and technical breaks [12], and harbouring a microbiome environment that could compensate for toxic build-up of trace gases in the growth chamber [11]. Moreover, inedible crop parts and food waste, as well as human waste, can be composted and added to the soil to provide nutrients for the next harvest. This adds a circularity factor to the system that would minimize, and eventually eliminate, any need for expensive resupplies of nutrients from Earth [13].

Gardening on Mars *is* viable. Although Mars has a very thin atmosphere (1% that of Earth), it contains the necessary elements for plant growth, such as carbon dioxide ($CO_2$) and nitrogen ($N_2$) [14], and the planet is abundant in (frozen) water present both in the polar caps and in several places under the Martian surface [15]. Because of its thin atmosphere, Mars has high

radiation levels on the surface which can be harmful to both plants and humans [16], and the planet holds an average global temperature of -65°C [17]. To counteract these conditions, colonies on Mars will likely consist of closed habitats that can shield both plants and humans from harmful radiation, and that can maintain the optimum temperature and atmospheric conditions for life within the colony [18]. Gravity on Mars is about 38% that of Earth [17]. However, experiments done on plant growth under microgravity conditions aboard the International Space Station (ISS) showed that plants, including crop species such as lettuce, cabbage and mustard, could be successfully grown, harvested and safely consumed by astronauts [19], showing promise that the lower gravity will not be a major impediment for normal plant development on Mars.

There are some potential obstacles with the Mars regolith that need to be overcome in order to successfully grow plants using *in situ* resources. For example, samples from Mars missions have shown the presence of heavy metals and perchlorates in the Martian regolith across different landing sites [20], harmful to both plants and humans. However, phytoremediation and bioremediation offer potential solutions to this, where specific plants or microbes can be cultivated to remove such elements from the soil with up to 100% efficacy [21–23]. Moreover, there's the suggestion that perchlorates would not be present in deeper layers of Martian regolith, since their formation on Mars is proposed to be largely due to the regolith interaction with cosmic radiation and other atmospheric processes [24]. Additional concerns include the regolith's high pH (~8.5) and nutrient deficiency (especially of nitrogen), both detrimental to plant growth and health. To address these, the addition of compost to the soil could lower the pH [25] while also providing the lacking nutrients. An initial batch of fertilizer could be incorporated in first missions' cargo supply, until a cyclic bioregenerative system could be established. Such system could use inedible parts from the crops grown *in situ* as compost for the next crop generation, or make use of human faeces as manure [26].

## Optimising resource use in Martian colonies

On Earth, it is possible to test the viability of using Martian soil (and thus of having a Martian soil-based system) for crop growth using Mars regolith simulants available. Several crop growth experiments have been successfully carried out on Earth using such simulants, with over 20 different species of crops reported to be able to grow and produce yield [27–29]. However, the main focus of these pioneering experiments was to test whether plants *could* germinate, grow and produce yield in the Mars regolith. No attention has yet been given to developing the *efficiency* of the system. Since colonies on Mars will probably operate under a limited amount of energy, space and other resources, any crop growth method that increases the productivity and circularity of the system, in a way that optimizes the use of such space and resources, will be paramount to the success and viability of the mission.

Intercropping is an ancient technique used on Earth that involves growing two or more crop species simultaneously in the same field [30]. This differs from the "traditional" monocropping system, where only a single crop species is planted in the same field. Intercropping has long been known to produce several beneficial effects compared to monocropping, such as increasing plant biomass and yield of one or more of the component species, improving the nutritional value of yield, and optimizing the use of resources in the system [31–33]. Such effects have been observed both in the field as well as in greenhouse pot experiments [34–36]. The beneficial effects of intercropping are often measured as a more efficient use of resources compared to the performance of each separate species in a monocropping system.

One way intercrop productivity benefits can arise is when the interspecific competition (competition between different species) in the intercropping system is *lower* than the

intraspecific competition (competition amongst the same species) [37]. Another way is by means of *species complementarity*, where both (or all) intercropped species can mutually benefit each other within the system. Complementarity can work either by 1) *niche differentiation*, where companion species that are adapted to different niches (i.e. a shallow root species together with a deep root species) can make a more complete use of all available resources in the system, resulting in less competition for resources; or by 2) *resource facilitation*, where one companion species provides or facilitates the up-take of a limiting resource to the other companion species [38]. A classic example of resource facilitation in intercropping is seen through legumes, which can provide nitrogen (N) into the intercropping system thanks to their symbiotic relationship with rhizobia bacteria. The bacteria infect the legume′s roots to form "root nodules", wherein they turn atmospheric nitrogen ($N_2$) into the usable form of ammonia ($NH_3$) via a process called N fixation [39]. This process helps relieve the competitive pressure on the intercropping system, either via the addition of N in soil [40] or via direct N transfer from the legume to the non-legume [41], and often results in the optimization of resources such as lower N fertilizer requirements [31].

The benefits of intercropping towards optimizing yield and resource use make it a prime candidate method to be applied on a soil-based system on Mars. To date, no experiments have ever been done to investigate the effects of intercropping on crops grown in Mars regolith. In this study, we aim to test the viability and prospect of implementing an intercropping system as a potential food production method in future Mars colonies.

## Study objective and research question

Our study objective is to investigate the effects of intercropping on crop performance grown in a Mars regolith simulant (from now on also referred to as just "Mars regolith"). This is an unprecedented and novel approach to tackling the challenge of optimizing resource use efficiency for soil-based food production in Martian colonies. We chose three companion species, namely pea (*Pisum sativum*), carrot (*Daucus carota*) and tomato (*Solanum lycopersicum*), for their complementary and nutritional properties, and for the fact that they have all been previously shown to be able to grow and produce yield in Mars regolith simulant [29]. We quantify plant performance as above-ground biomass (g), yield (g), harvest index (%), and nutritional content of yield as levels of nitrogen (N), phosphorus (P) and potassium (K) (g/kg). The overall intercropping system performance is also quantified using the Relative Yield Total (RYT) index, where a value of higher than 1 indicates an overall yield advantage between all species in the intercropping system—compared to their monocropped counterparts—and a value of less than 1 indicates an overall yield disadvantage. Nodulation by rhizobia bacteria in pea roots will also be a key variable observed in this experiment, to interpret the effects that an N-fixing species would have on the interspecific competition within the intercropping system.

Our research question was whether we would see any beneficial effects of intercropping on the performance of pea, carrot and tomato grown in Mars regolith simulant. We hypothesise that, for all soil types, all three species grown in the intercropping system will give higher values for each of the plant performance indicators, compared to their monocropped counterparts. In turn, we also hypothesise that the intercropping system will show an overall yield advantage when compared to monocropping, giving an RYT value of higher than 1, for all three soil types. If plants perform better when intercropped, by making use of the exact same resource input as when they are monocropped, this could prove the efficacy of intercropping as a method for increasing the efficiency of the system as a whole.

## Materials and methods

### Greenhouse conditions

We performed a pot experiment in a greenhouse on Earth, in Wageningen (51.9692° N, 5.6654° E). During the experimental period, average temperature in the greenhouse was 20.4 ± 1.7°C, relative humidity was 60.9 ± 15%, and day/night cycle was 16/8h. Ambient air was used and no extra $CO_2$ was added. Lamps yielding 600 watt (HPS 230 volt) were switched on if sunlight intensity was below 150 watt/m$^2$, and switched off if sunlight intensity was above 250 watt/m$^2$.

### Mars regolith simulant

The regolith analogue used in this experiment was the MMS-1 Mars Regolith Simulant, of unsorted grade (grain <3.17mm, density 1.25g/cm$^3$), provided by The Martian Garden Company (Austin, USA). This simulant was manufactured under commission by researchers at the National Aeronautics and Space Administration (NASA)'s Jet Propulsion Laboratory (JPL) [42]. MMS-1 was designed to be the closest simulate of the regolith found on the Martian surface, especially when in contact with water. For a comparison of physical and chemical properties of the MMS-1 simulant and actual Martian regolith see S1 Appendix.

### Earth soils and soil preparation

Two "Earth" soils were included in the experimental design: (1) common potting soil, composed mainly of peat with fertilizer (PGmix 15%N-10%P2O5-20%K2O), and (2) river sand. The potting soil was sieved to remove lumpy matter and improve homogeneity. Sand was specially chosen for it being nutrient poor, similar to Mars regolith (see also [28], to test if they would produce comparable results.

For the Mars regolith and the sand treatments, a small amount of organic soil (derived from the sieved potting soil) was mixed at 10% of the total volume, to improve root aggregation and water retention [43]. Table 1 shows the pH and Nutrient analysis from samples of the three soils.

### Species selection

The species selected for this experiment were pea (*Pisum sativum* cv. 'Prince Albert', Tuin Plus), carrot (*Daucus carota* cv. 'Nantes 2', Buzzy), and dwarf cherry tomato (*Solanum lycopersicum* cv. 'Tiny Tim', Buzzy). These species represent 3 out of the 27 potential food candidates to be incorporated into a space habitat, as proposed in NASA's Life Support Values and Assumptions Document [44], based on various selection criteria such as potential yield, harvest index, horticultural requirements and macronutrient content needed to supplement a crew's diet.

**Table 1. Nutrient composition and pH for the three soil treatments (MMS-1 Mars regolith simulant, sand and potting soil), collected at the start of the experiment (day 0).** Results shown for Mars regolith simulant and sand include a 10% volume of organic matter mixed with each.

| Nutrient (mg/kg) | Mars regolith simulant (MMS-1) | Sand | Potting soil |
|---|---|---|---|
| N-NH4 | 4,34 | 3,71 | 343,16 |
| P-PO4 | 0,1 | 0,1 | 41,77 |
| N-NO3 | 15,94 | 5,27 | 180,70 |
| Nmin | 20,3 | 9 | 523,9 |
| pH | 8.5 ± 0.2 | 7.5 ± 0.2 | 5.5 ± 0.2 |

These species have also been chosen for their particular complementary properties which can be mutually beneficial to each other. As a legume, pea can serve as N-fixers in the system in symbiosis with rhizobia bacteria. Carrot helps aerate the soil, which can improve water and nutrient uptake by companion plants [45]. Tomato can provide shade for the temperature sensitive carrot and support for the climbing pea [46], and both carrot and tomato release root exudates such as flavonoids that can promote root nodule formation in pea [47]. Furthermore, all three species have compostable crop waste that can be mixed with the soil to provide key nutrients to the subsequent crop, reducing the need for fertilizers and promoting circularity in the system [48].

Finally, these species were chosen for their nutrient composition. All three species, in particular carrot and tomato, are high in antioxidants such as vitamin C and β-carotene [49–51]. These are an essential part of the human diet, but most importantly are amongst the few key nutrients that are mostly or completely lost in dehydrated foods [4].

### Rhizobia bacteria addition

To provide the peas with their symbionts for N-fixing, 0,75ml of *Rhizobium leguminosarum* (biovar *viciae* strain 248 OD600 ~ 0.784) was added to every pot, including pots not containing pea plants. Inoculation was done 19 days after sowing. The selection of *R. leguminosarum* was based on its well-documented symbiotic relationship with *P. sativum* [52]. This particular strain was chosen due to its prior successful nodulation with *P. sativum* in a previous in-house experiment, where the same Mars regolith simulant (MMS-1) was used, thus aiming to control the potential variable of the bacteria not adapting to Martian regolith conditions.

### Experimental design

We used a randomised complete block design with the three experimental species grown in three types of soil (MMS-1 Mars regolith simulant, sand and potting soil), divided into two cropping systems (intercropping and monocropping). This gave 12 different treatments with five replicas for each treatment, resulting in 60 pots. The experimental design and treatments can be seen in Fig 1.

For the monocropping treatments, three plants of the same species were grown in each pot, whilst for the intercropping treatments, one plant of each species was grown in each pot. 5L Pots were used, 16cm deep and 20cm wide at the top. Pots were filled to a 13cm depth, with 4.55Kg Mars regolith, 5Kg sand and 1.3Kg potting soil. A mesh was placed on the bottom of each pot to keep the soil in place.

Seeds were sown on 1st December 2021 and the experiment ran for 105 days, with the harvest taking place on 16th March 2022. Seeds were over-sown to ensure the required germination. Thinning of germinated seeds was carried out 10–14 days later.

### Watering and nutrient solution

Water saturation point for each soil was measured before the start of the experiment (Mars regolith = 35.7%, potting soil = 119%, sand = 27%). Watering was done once a day, every day, by pouring water on the saucer beneath the pot. For this, a random sample of three pots from each soil type was weighed and the amount of water to be given was determined to be just below the saturation point for that soil.

100ml of 2EC Hoagland nutrient solution was given once a week for the first 68 days, then increased to twice a week for the remainder of days to compensate for the increased biomass from the growing plants. The solution was poured on the topsoil surface. Solution contents can be found in S2 Appendix.

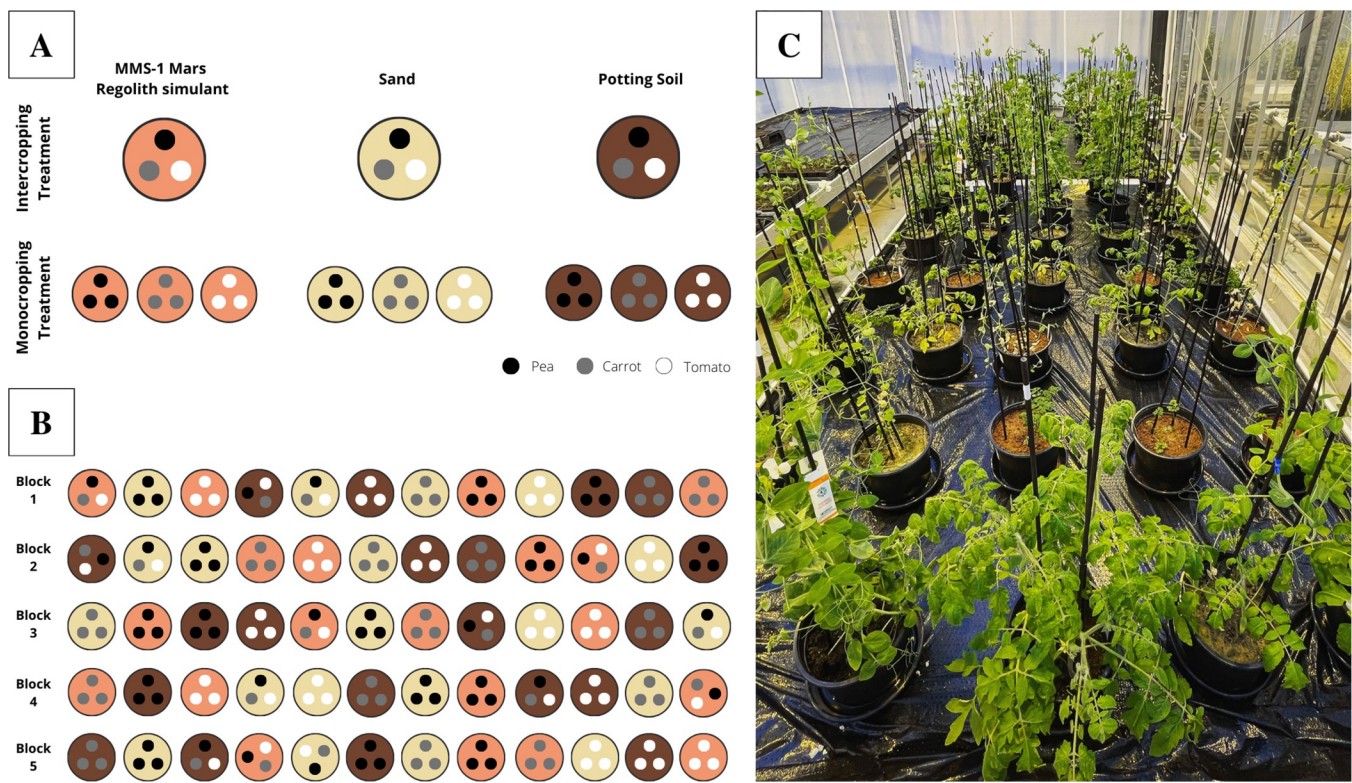

**Fig 1. Experimental design.** A: Overview of experimental treatments. B: Randomised complete block design with five replicas, and intercropping treatment pots randomly rotated. C: Photo of the experiment (day 48).

### Harvesting and measurements

At harvest, above-ground biomass and yield were weighed for all three species. Both variables were measured in grams per plant (for monocropping treatments these values were added and divided by the number of plants in the pot, so to get the average value per pot). Above-ground biomass included stem, leaves and branches. Yield fresh weight was done separately for ripe and immature tomato fruits, and for pea pods and pea seeds, where only the seeds were considered for yield analysis. Harvesting of pea yield was also done throughout the experiment, as the first pea seeds started to mature on day 65. The first tomato fruits only started to ripen two weeks before the harvest date, so they were left on the plants to be harvested all at once. For carrot yield, the whole carrot from the crown to the tap root was weighed. All pea roots were washed and visually assessed for the presence of rhizobia nodules, which were also dissected and checked for nodule activity using a light microscope (active nodules show an inner pink coloration [53])

All fresh weighed samples were put in ovens to dry at 70°C for 48 hours.

Yield of the three species was analysed for NPK content. Three replicas of each treatment were grinded and analysed using digestion H2SO4/H2O2/Se + NP on SFA and H2SO4/H2O2/Se + K on ICP-OES. For tomato, only ripe (red) tomato fruits were analysed.

### Calculations

Harvest index (HI) is a measure of the reproductive efficiency, indicating how much of a plant's above-ground biomass is allocated to yield. It was calculated from dry weight data, for

all species, according to Eq 1:

$$HI(\%) = \frac{Y}{Y + ABG} * 100 \qquad (1)$$

Where Y is the yield (g), and ABG is the above ground biomass (g).

The division of the intercropping yield over the monocropping yield of the same species gives the relative yield (RY) for that species (Eq 2). A RY of > 0.33 (0.33 being the ratio of each species present in the intercropping treatment) indicates that the species over-yielded in the intercropping treatment compared to its monocropping treatment.

$$RY = \frac{Y_i}{Y_m} \qquad (2)$$

Where $Y_i$ is the yield from the intercropping treatment (per species) (g), and $Y_m$ is the yield from the monocropping treatment (averaged between the number of plants in the pot) (g).

Relative yield total (RYT) is an index to assess whether there was any yield advantage in the intercropping system compared to monocropping. It is a useful measure to assess how efficiently the intercropping system performed as a whole [54]. It was calculated according to Eq 3:

$$RYT = RYt + RYp + RYc \qquad (3)$$

Where RYt, RYp and RYc are the relative yields of tomato, pea and carrot (respectively). If RYT > 1, this indicates relative yield (and hence resource use) advantage from intercropping, and RYT < 1 indicates relative yield disadvantage from intercropping.

## Data analysis

All data was checked for normality and homogeneity using a W test for normality and a Barlett's test for homogeneity. When needed, $LOG_{10}$ transformations were done to meet the two assumptions, and tests were then run on the transformed data. There were only three parameters that did not fit homogeneity, tomato yield P content (P = 0.024), and harvest index for peas and for carrots (P = 0.011), but those were still included in the analysis.

A two-way ANOVA was performed on the following variables: above-ground biomass dry weight, yield dry weight, harvest index, N content in yield (NY), P content in yield (PY) and K content in yield (KY). For tomato yield, the data also included the dry weight of the immature fruits harvested, since they would become ripe and consumable if harvested later. The ANOVA was run separately for each species, with cropping type and soil type at the interaction level, and block effects accounted for. A post-hoc Fisher's protected LSD test (P<0.05) was performed for significant effects of soil type and the interaction between soil type and cropping type.

An analysis of deviance was performed on the ratios between intercropping and monocropping for each species in every soil (calculated from LOG-transformed data), and the p-values were obtained for the null hypothesis that the ratio equals 1 (i.e. no difference between intercropping and monocropping treatments).

There was a missing value in one of the replicas sent for the NPK analysis of pea yield from the intercropping Mars regolith treatment due to insufficient pea material available. This means that for this treatment, only 2 replicas were analysed for NPK. There were no other missing or deleted values in the data set.

## Results

### Mars regolith

Tomato showed increased performance in the intercropping treatment compared to the monocropping treatment, with significantly higher above-ground (AG) biomass (P = 0.037), yield (P = 0.043), and Potassium content in yield (KY) (P<0.001, Table 2, Figs 2 and 3).

Contrary to tomato, carrot showed decreased performance in the intercropping treatment compared to the monocropping treatment, with significantly lower AG biomass (P = 0.023), yield (P = 0.022), NY (P<0.001) and PY (P = 0.011, Table 2).

Pea showed no significant difference in performance between the intercropping and monocropping treatments for any of the variables. However, like carrot, the observed trend showed lower values in the intercropping treatments compared to the monocropping treatments for all variables, with the p-value for N content close to the threshold (P = 0.060, Table 2).

There was no significant difference in harvest index between intercropping and monocropping for any of the three species.

### Comparison between soils

Potting soil produced the same pattern of results as those seen in the Mars regolith simulant, when comparing the intercropping and monocropping treatments (Table 2 and Fig 3). The only exception was KY, where in tomato it showed no significant difference, and in pea it was significantly lower in intercropping compared to monocropping (P = 0.029).

Sand also produced a similar pattern of results to Mars regolith, when comparing the intercropping and monocropping treatments (Table 2 and Fig 3), with three exceptions where the values were inverted: Tomato showed no significant difference in KY, but showed an indication of *lower* K in intercropping; Pea showed significantly *higher* AG biomass (P = 0.002) and yield (P = 0.012) in intercropping compared to monocropping; and carrot showed no difference in NY, but with an indication of *higher* N in intercropping. A last exception was carrot PY, where there was no significant difference between cropping treatments, but the pattern followed being lower for intercropping compared to monocropping.

For all three species, AG biomass, yield, and harvest index mostly followed the pattern of being significantly the highest on potting soil, followed by sand, and lastly by Mars regolith (P<0.001). Where the difference was not significant (P>0.05), the values always followed the same pattern, except in the harvest indexes of tomato intercrop and monocrop, pea intercrop and carrot monocrop, where sand values were slightly higher than in potting soil (Fig 3A–3C). The only significant exception to this pattern were carrot AG biomass and yield, where both were lower in the potting soil intercropping treatment compared to the sand monocropping treatment (P<0.001, Figs 4 and 5). A noteworthy change was seen in pea AG biomass and yield, where both went from being significantly lower in sand compared to potting soil in monocropping, to having no significant difference between the two soils in intercropping (AG biomass: P = 0.030; yield: P = 0.015) (Fig 3A and 3B).

For NPK content in yield, the pattern seen above was inverted, where Mars regolith was mostly found to give the highest yield NPK values, and potting soil the lowest values, for both cropping treatments. Mars regolith had significantly higher values than both sand and potting soil for tomato NY and KY (P<0.001), pea P (P = 0.007) and K (P<0.001), and carrot N (P<0.011). No significant difference was found amongst any other treatments. The only exception to this pattern was tomato PY, where it was significantly higher in Potting soil compared to Mars regolith monocrop, and higher in sand monocrop compared to Mars regolith monocrop (P = 0.041, Fig 3D–3F).

**Table 2. Average values (N = 5) of all six measured variables for tomato, pea and carrot for the intercropping and monocropping treatments, for each of the three soils (MMS-1 Mars regolith simulant, sand and potting soil).**

| | Tomato | | Pea | | Carrot | |
|---|---|---|---|---|---|---|
| | Intercropping | Monocropping | Intercropping | Monocropping | Intercropping | Monocropping |
| **AG[1] Biomass (g)[2]** | | | | | | |
| Mars regolith | 4.42 (±0.23)** a | 2.15 (±0.35) a | 0.45 (±0.10) a | 1.01 (±0.14) a | 0.19 (±0.08)** a | 0.60 (±0.13) ab |
| Sand | 6.76 (±3.92) * b | 4.45 (±2.00) b | 7.41 (±1.59)** c | 4.01 (±0.74) b | 1.07 (±0.06)** b | 1.71 (±0.30) c |
| Potting soil | 24.71 (±1.26)** c | 15.68 (±0.28) c | 7.72 (±1.11) c | 8.18 (±0.88) c | 0.79 (±0.26)*** b | 4.62 (±0.15) d |
| **Yield (g)[2]** | | | | | | |
| Mars regolith | 2.47 (±0.41)** ab | 1.29 (±0.37) a | 0.25 (±0.14) a | 0.46 (±0.08) a | 0.36 (±0.15)** a | 1.12 (±0.19) a |
| Sand | 5.32 (±0.56)** c | 3.26 (±0.45) b | 8.82 (±1.89)** c | 4.71 (±0.94) b | 2.89 (±0.38)*** b | 6.46 (±0.37) c |
| Potting soil | 17.18 (±0.76)*** e | 8.68 (±0.40) d | 8.72 (±0.61) c | 10.44 (±1.30) c | 2.26 (±0.53)*** b | 12.04 (±0.43) d |
| **Harvest index (%)** | | | | | | |
| Mars regolith | 34.84 (±3.44) a | 34.98 (±3.82) a | 22.34 (±10.20) b | 30.93 (±4.79) b | 63.92 (±8.17) a | 65.75 (±2.74) a |
| Sand | 45.85 (±5.31) a | 41.73 (±3.24) a | 55.24 (±1.20) c | 50.17 (±1.65) a | 73.80 (±2.73) b | 78.98 (±1.90) b |
| Potting soil | 42.80 (±3.41) a | 36.66 (±3.22) a | 52.74 (±2.86) a | 56.07 (±8.29) a | 73.04 (±1.62) ab | 72.25 (±1.91) ab |
| **N content in yield (g/kg)** | | | | | | |
| Mars regolith | 21.93 (±1.16) b | 18.47 (±3.51) b | 28.35 (±5.10)* a | 39.63 (±2.26) a | 9.73 (±0.88)*** c | 16.03 (±1.64) d |
| Sand | 12.57 (±0.39) a | 10.80 (±0.32) a | 37.60 (±1.63) a | 35.97 (±3.08) a | 7.50 (±0.95) abc | 6.80 (±0.62) ab |
| Potting soil | 10.87 (±1.01) a | 10.13 (±0.34) a | 35.20 (±2.43) a | 33.80 (±2.33) a | 5.13 (±0.74)** a | 8.27 (±0.24) bc |
| **P content in yield (g/kg)** | | | | | | |
| Mars regolith | 3.96 (±0.37)* ab | 3.41 (±0.25) a | 4.15 (±0.96) b | 5.01 (±0.24) b | 1.58 (±0.11)** a | 2.74 (±0.22) a |
| Sand | 3.95 (±0.11)* ab | 4.58 (±0.05) b | 2.90 (±0.11) a | 3.25 (±0.54) a | 2.45 (±0.32) a | 3.01 (±0.14) a |
| Potting soil | 4.40 (±0.04) b | 4.36 (±0.11)b | 2.90 (±0.20) a | 3.75 (±0.62) a | 2.30 (±0.52)** a | 3.31 (±0.17) a |
| **K content in yield (g/kg)** | | | | | | |
| Mars regolith | 55.00 (±3.29)*** c | 40.87 (±0.44) b | 14.50 (±0.49) b | 14.60 (±0.25) b | 18.13 (±3.94) a | 22.93 (±1.26) a |
| Sand | 35.90 (±1.36)* a | 39.70 (±0.38) ab | 12.10 (±0.20) a | 13.00 (±0.57) a | 21.67 (±2.50) a | 24.27 (±2.09) a |
| Potting soil | 37.87 (±1.58) ab | 35.93 (±0.49) a | 12.47 (±0.32) ** a | 13.77 (±0.49) a | 15.60 (±4.74)* a | 24.97 (±2.45) a |

1. "AG" = Above-ground.

2. Values refer to dry weight.

Values in brackets refer to standard error (SE).

Asterisks (*) refer to significant differences between intercropping and monocropping treatments *within* each species from the *same* soil treatment, at the 0.1 level (*), 0.05 level (**) and 0.001 level (***) (Analysis of Deviance).

Letters refer to differences between *soil* treatments within each *variable* for each species. Different letters indicate a significant difference at the 0.05 level (Fisher's LSD test).

Sand was the only soil to present an overall yield advantage in the intercropping system compared to monocropping (RYT = 1,32, Table 3 and Fig 6), where both tomato and pea showed yield advantage in intercropping over monocropping (tomato relative yield (RY) = 0.54; pea RY = 0,62), and carrot showed yield disadvantage (RY = 0,15). Potting soil showed no overall yield advantage (RYT = 1,00), where tomato over-yielded (RY = 0,66) but both pea and carrot under-yielded (pea RY = 0,28; carrot RY = 0,06). Mars regolith showed under-yielding in the intercropping system compared to monocropping (RYT = 0,93), where tomato over-yielded (RY = 0,64) but both pea and carrot under-yielded (pea RY = 0,18; carrot RY = 0,11).

## Presence of rhizobia bacteria

Visual analysis of roots of peas in both monocropping and intercropping treatments revealed presence of Rhizobia infected nodules in sand and potting soil, but not in Mars regolith (except

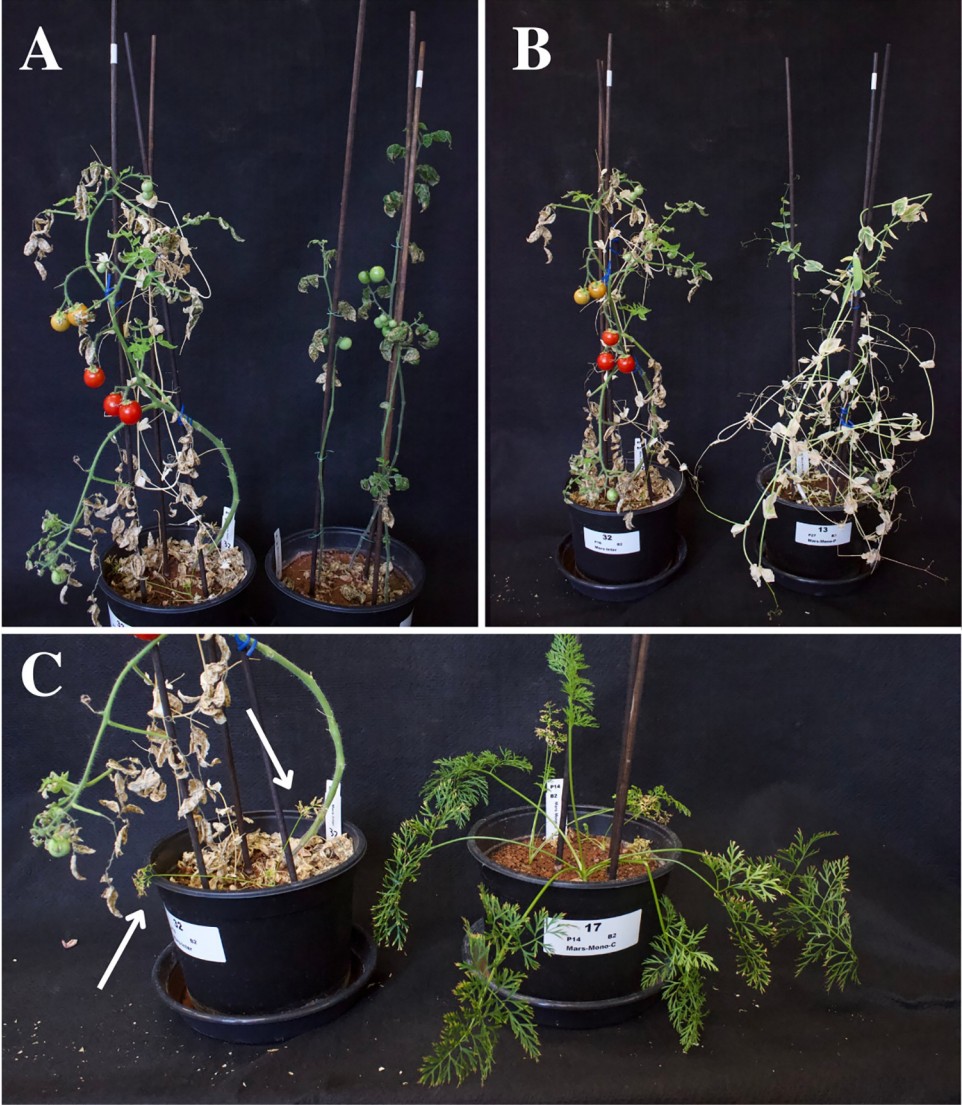

**Fig 2. Comparison between intercropping and monocropping treatments in Mars regolith simulant.** A: Intercropping treatment (left) and monocropped tomato (right). B: Intercropping treatment (left) and monocropped pea (right). C: Intercropping treatment (left) and monocropped carrot (right). Arrows point to the small carrot leaves from the intercropping treatment. Pictures were taken on the day of harvest (day 105).

for one replica from a monocropping treatment which had 3 nodules). Between 17 and 150 nodules were found across the other treatments. There was no significant difference in the number of nodules between potting soil and sand (Fisher's post hoc test P<0.05), although there was on average more nodules found in sand than in potting soil (Fig 7). Both sand and potting soil showed a significantly higher number of root nodules than Mars regolith (P<0.001). There was also on average more nodules in intercropping compared to monocropping for both soils, although there was considerable variation between replicas, and thus this difference was not significant (P = 0.100). Dissection of several nodules per replica showed a pink coloration inside, indicating that the nodules were active (Fig 8).

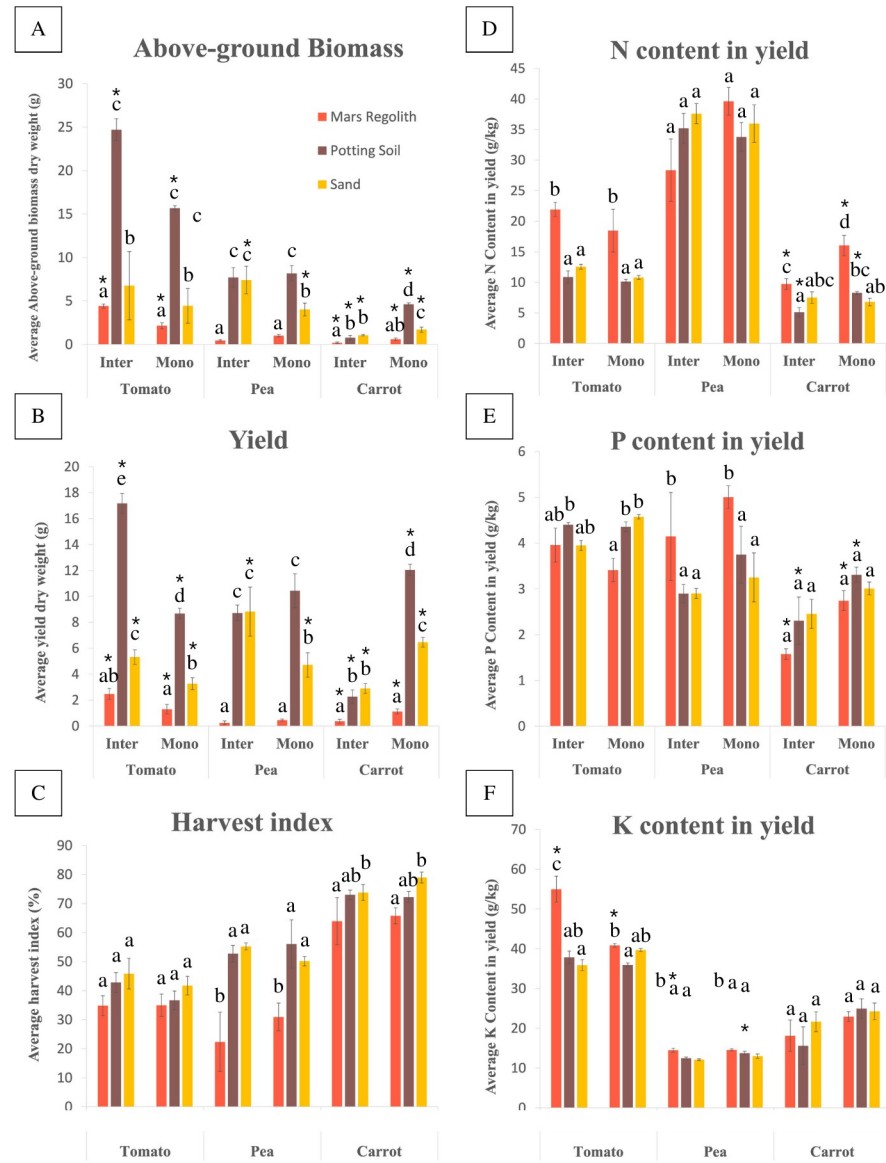

**Fig 3. Comparison between intercropping and monocropping for tomato, pea and carrot in all three soils (Mars regolith simulant, potting soil and sand).** A: Above-ground biomass. B: Yield. C: Harvest index. D: N content in yield (NY). E: P content in yield (PY). F: K content in yield (KY). "Inter" = Intercropping treatment, "Mono" = Monocropping treatment. Letters indicate significant differences between soils within the same cropping treatment and within the same species, at the 0.05 level (Fisher's LSD test). Asterisks (*) indicate significant differences between intercropping and monocropping treatments within the same soil and within the same species, at the 0.05 level (Analysis of Deviance). Error bars represent standard error (SE).

## Discussion

### Intercropping effects on RYT for Mars regolith

Overall, the intercropping system in Mars regolith showed yield disadvantage over monocropping, with an RYT (total relative yield) value of 0.93. Intercropping advantages or disadvantages can indicate the amount of interspecific competition or facilitation that is occurring within an intercropping system [54]. An RYT of less than 1 means that interspecific

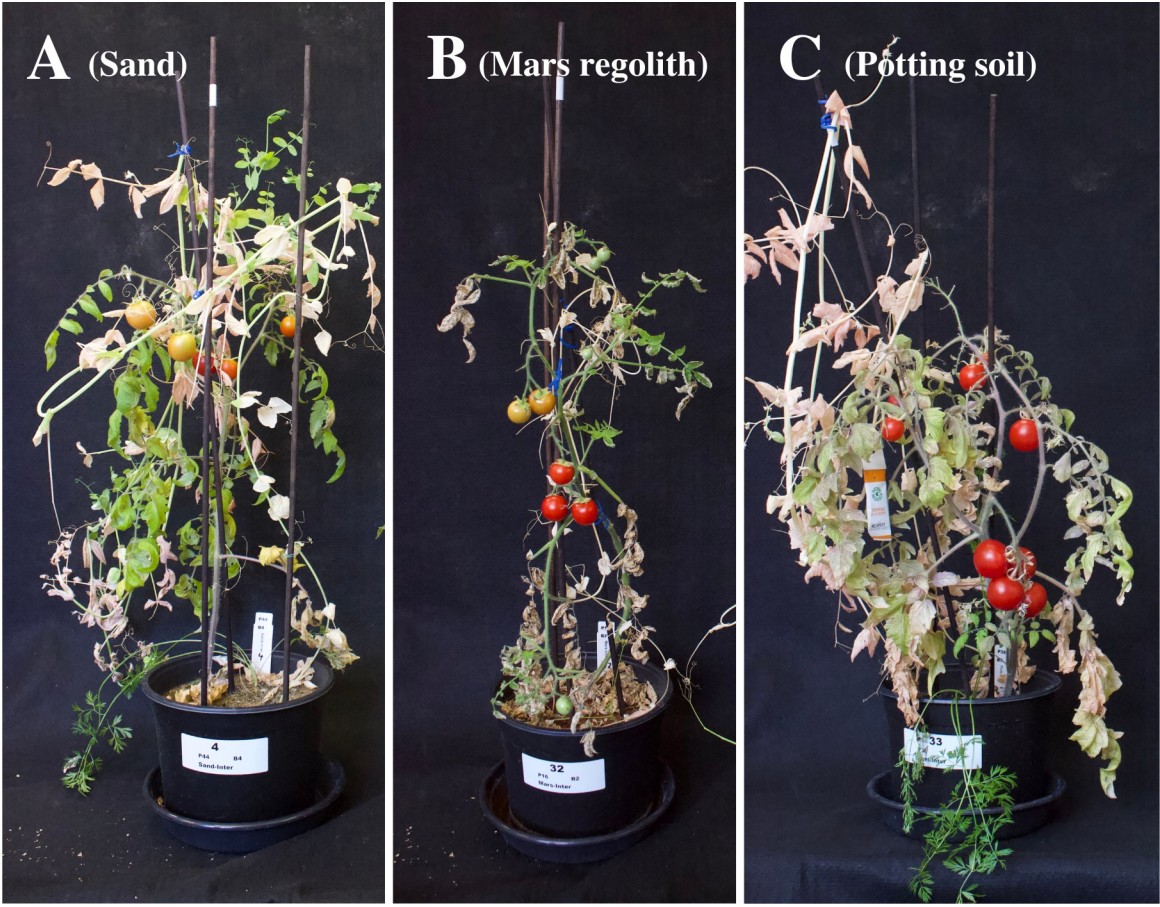

**Fig 4. Comparison between intercropping treatment from the three soils.** A: Sand. B: Mars regolith simulant. C: Potting soil.

competition in intercropping was higher than intraspecific competition in monocropping. This is very likely due to the fact that pea did not form root nodules in symbiosis with the Rhizobia in Mars regolith, thus negating its role as N-fixers and the key advantage of having a legume in an intercropping system. Many studies have demonstrated that when intercropped, pea can compensate for increased nutrient competition, as well as for N deficiency in the soil, by releasing root exudates (i.e. flavonoids) that can promote root nodule formation [47,55]. This in turn increases fixation of atmospheric N and helps pea both escape the competitive pressure, as well as relieve the pressure for the other species by leaving more N available in the soil. Without the formation of nodules this was not possible, and instead of relieving the competitive pressure, pea became an added competitive load to the intercropping system in Mars regolith.

The contrast in performance between tomato and the other two species can be attributed to the fact that different crops often show different levels of competition strength when intercropped together. Tomato is a dominant species within intercropping systems, and has been shown to have yield advantages when intercropped with legumes [56]. In another study by Wu et al. [36], where tomatoes were intercropped with potato onion, intercropping promoted the growth of tomato, whilst it inhibited the growth of potato onion.

Tomato is also generally known to be "heavy feeders" with a high demand for nutrient supply [57], which could mean that they monopolise the resource uptake in the pot. This higher

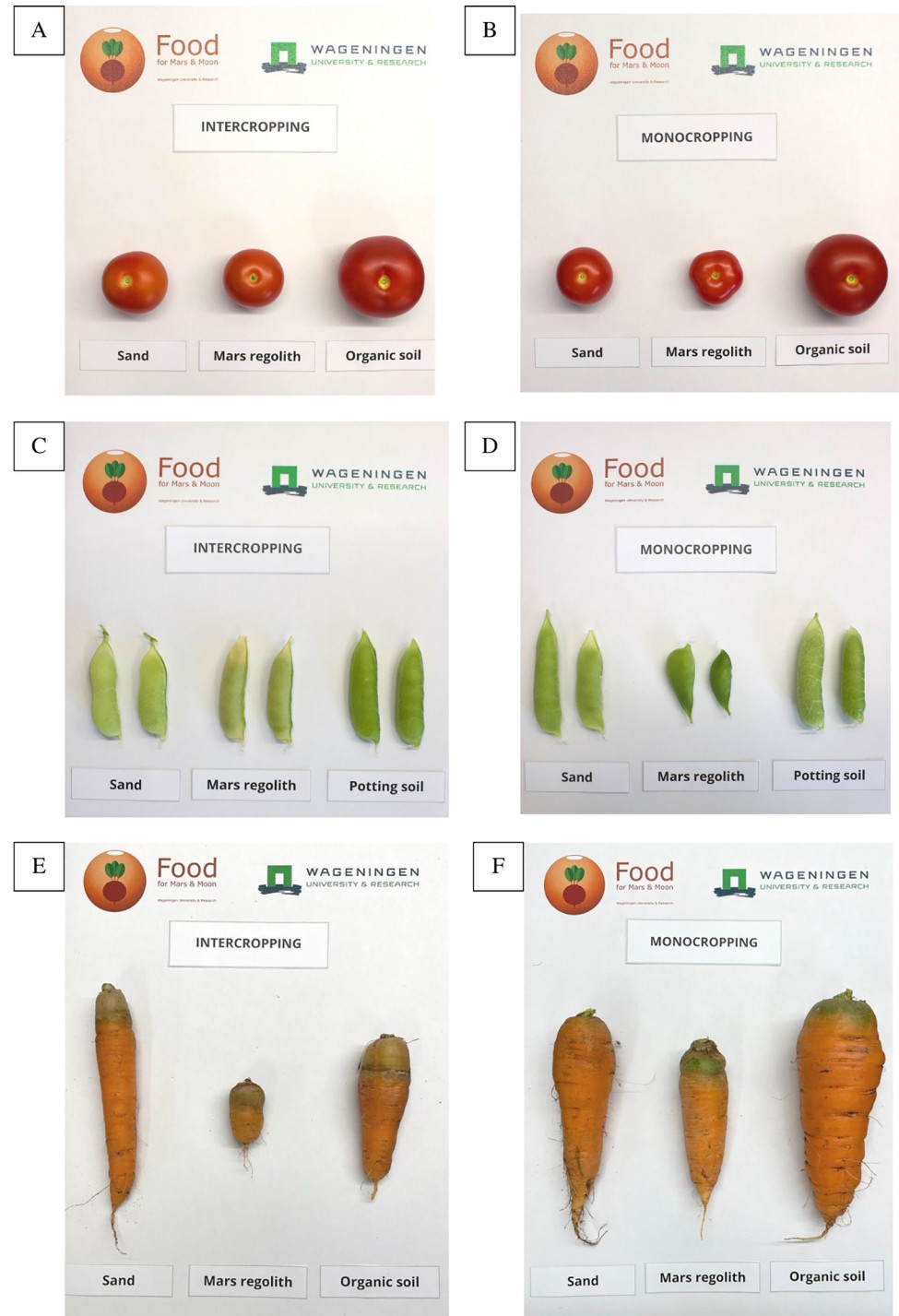

**Fig 5. Yield comparison between cropping treatments and between soils.** Labels in pictures indicating "Organic soil" refer to the potting soil treatment. A: Intercropped tomato. B: Monocropped tomato. C: Intercropped peas. D: Monocropped peas. E: Intercropped carrots. F: Intercropped carrots.

competitive strength of tomato could explain why it had increased yield performance in intercropping compared to monocropping, since it was put at an advantage compared to pea and carrot, whilst losing this advantage when grown with other tomato plants. It could also explain

**Table 3. Relative yields for tomato, pea and carrot in each soil treatment, and the total relative yield (RYT) for the whole intercropping system when compared to monocropping, for each of the three soils (MMS-1 Mars regolith simulant, sand and potting soil).**

| Soil | Relative yield | | | RYT |
|---|---|---|---|---|
| | Tomato | Pea | Carrot | |
| **Mars regolith** | 0,64 | 0,18 | 0,11 | 0,93 |
| **Sand** | 0,54 | 0,62 | 0,15 | 1,32 |
| **Potting soil** | 0,66 | 0,28 | 0,06 | 1,00 |

why in Mars regolith pea and carrot had lower performance in the intercropping treatment, as a higher resource uptake would confer the tomato more nutrients for growth, whilst leaving less resources available to the other two species. Carrot, which depends solely on the nutrients available in the soil, would naturally be at a disadvantage when grown with tomato.

Interestingly, in sand where we saw an adequate number of nodules in pea roots ("adequate" meaning an average of 30 or more nodules per plant [58]), there *was* an overall yield advantage in the intercropping system (RYT = 1.32), where although intercropped carrots underperformed, both pea and tomato showed significantly higher yield compared to monocropping. This higher RYT owes to the fact that pea performed particularly well when intercropped in sand, giving equal or higher values in all variables compared to monocropping.

The success of peas in sand is most likely due to the higher number of root nodules that were formed, particularly in the intercropping treatment, allowing for more atmospheric N-fixing. The higher number of root nodules in the sand treatment compared to potting soil can probably be explained by the lower N content available in sand, a factor that has been shown to promote nodulation [55]. Lower N availability has also been linked to an increase in RYT before: in a pea-barley intercrop study, RYT was highest for treatments with no fertilizer compared to treatments with applied fertilizer [59]. The high resource demand from the aggressive tomato in the sand intercropping treatment probably caused further decrease in N availability in the soil, which could explain the higher nodulation seen in this treatment (Fig 7).

## Potential factors affecting yield performance on Mars regolith

There are a few reasons why nodulation in pea may have been hindered in both Mars treatments. Many studies have shown nodulation and *R. leguminosarum* survival to greatly decrease when under a variety of biotic and abiotic stresses, including high salinity [60] and soil sterility [61], both of which are features of the MMS-1 Mars regolith used in this study. A potentially aggravating condition was that the regolith was very compact in comparison to both potting soil and sand. Although it was of "unsorted grade", it was composed mostly of very fine particles that gave it a clay-like texture when wet, and it did not drain well. The roots of the peas in Mars regolith reflected this in that they were smaller, thicker and more tortuous than the roots of peas in sand (and also in potting soil) (Fig 8), which is a common response to denser and coarser soils [62]. Since soil texture is known to affect salinity, where clay and moist soils tend to be more saline [63], this could have added a salt stress factor to the Mars regolith, negatively affecting rhizobia bacteria. Moreover, the Mars regolith was low in organic matter. Sterilised soils have been shown to severely hamper formation of nodules, when compared to unsterilised soils containing more varied microbial communities [61]. The chemical composition of the Martian regolith has also been suggested to affect nodulation. A study involving legume-Rhizobia symbiosis in Mars regolith simulants showed more nodulation in the MMS-2 regolith simulant compared to MMS-1, where MMS-2 contains added compounds such as Iron Oxide, Magnesium Oxide, Sulfates and Silicates [64]. Another similar study on

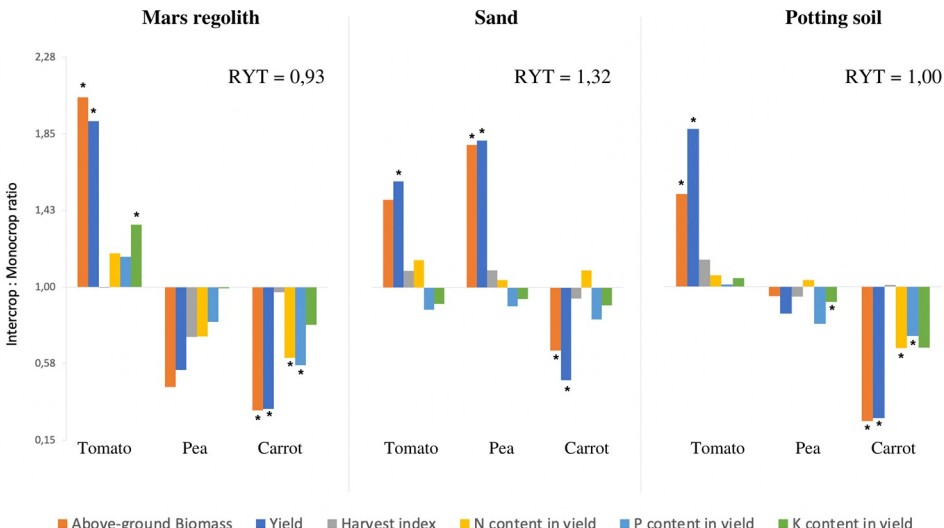

**Fig 6. Comparison of intercropping to monocropping ratios and overall system performance for the three species in each of the three soils (Mars regolith simulant, sand and potting soil).** Ratios above 1 indicate the variable had a higher value in intercropping relative to monocropping, ratios below 1 indicate the variable had a lower value in intercropping relative to monocropping. RYT (total relative yield) refers to overall yield advantage or disadvantage in the intercropping system. If RYT > 1, this indicates yield advantage from intercropping over monocropping, RYT < 1 indicates yield disadvantage from intercropping over monocropping, and RYT = 1 indicates no difference in yield between the two cropping systems. Variables indicated are above-ground biomass dry weight (g), yield dry weight (g), harvest index (%), and N, P and K content in yield (g/kg)) for each of the three species (tomato, pea and carrot). Asterisks (*) indicate significant differences between intercropping and monocropping values within the same variable, at the 0.05 level.

legume-Rhizobia interactions in MMS-1 suggested that the high pH of the regolith may impair Fe uptake by the plants, Fe being a key factor for plants to sustain a healthy relationship with rhizobia, which would in turn negatively impact nodulation [65].

The compactness factor of the regolith could have also posed a problem for plants themselves, regardless of how it may or may not have affected Rhizobia nodulation. Compact soils allow for lower gas diffusion and water conductivity, which can lead to anaerobic conditions in the soil and significantly reduce N availability and nutrient uptake [62], with more compact soils being linked to lower yields [66]. The organic matter present in Earth soils greatly contributes to soil quality, both in reducing soil compactness, as well as being rich in nutrients and the micro-organisms that play a part in making nutrients available for plant uptake [67]. This could also explain why the absolute values for yield were the highest in potting soil, and why sand also showed higher yield values compared to Mars (although sand is also poor in organic matter, it is not as sterile as the Mars regolith, and may have harboured a wider microbiome that aided in plant performance). The sterile nature of Mars regolith means that, although it has the *chemical elements* necessary to meet the requirements of plant growth (Table 1), these elements could be mostly absent in the bio-available forms that are necessary for their uptake [68], which are often made available by the activity of micro-organisms in the soil.

## Nutrient analysis

Potassium content in yield (KY) for Mars regolith intercropped tomato was significantly higher than KY for both sand and potting soil tomato, from both cropping treatments. This could be explained by the fact that optimum K uptake occurs only above 6.5 pH, and is

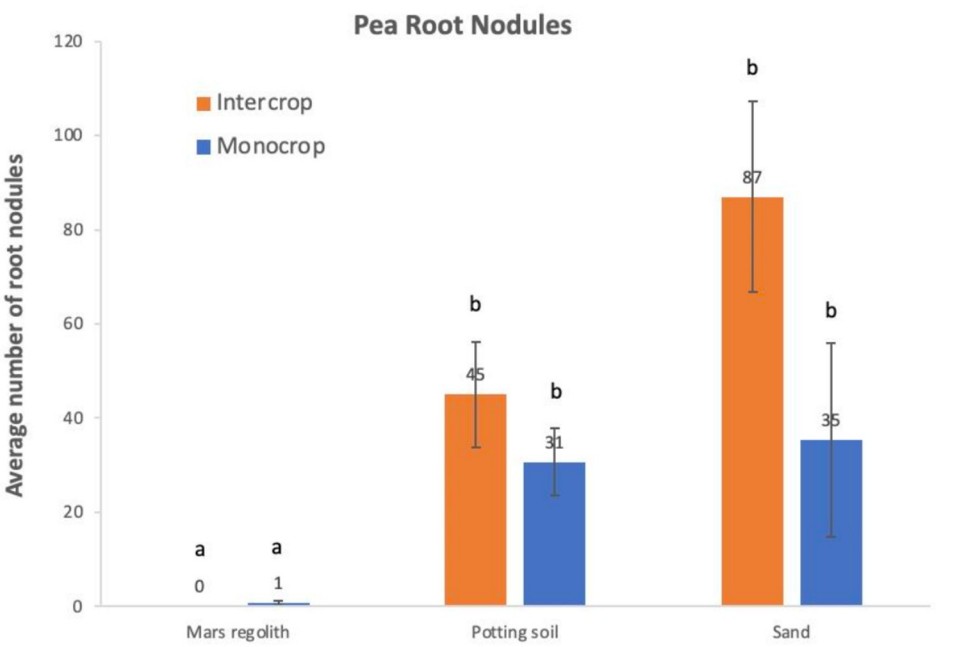

**Fig 7. Average number of root nodules on pea roots (N = 5) per plant.** Values are indicated for the monocropping and intercropping treatments from each of the three soils (Mars regolith, potting soil and sand). Data labels indicate the average value. Letters indicate significant differences between cropping treatment and between soils, at the 0.05 level (Fisher's LSD test). Error bars represent standard error (SE).

increased further with increased soil moisture [69], giving Mars regolith an advantageous condition for K uptake especially when compared to potting soil.

PY in Mars monocropped tomato was significantly lower than in Mars intercropped tomato. This could be explained by the fact that P uptake is severely compromised between pH 8 and pH 9, which is exactly where the Mars regolith falls within. However, it is unclear why this was also not the case for carrot and pea PY from Mars regolith, or for the Mars intercropped tomato treatment.

NY for both tomato and carrot were significantly higher in Mars regolith compared to potting soil, even though potting soil had vastly more nitrogen available (Table 1). Although higher N availability in soil has been linked to higher yields [70], studies have also shown that this higher availability has either no correlation or a negative correlation [71] with N content in yield. This could explain why in potting soil we see the highest values for yield but also the lowest values for N content in yield, and vice-versa for Mars regolith.

## Mars regolith limitations and proposed soil ameliorations

Physical and chemical properties of the Mars regolith, such as soil compactness and high pH, may have made it a hostile environment for survival and nodulation of rhizobia bacteria, as well as causing reduced nutrient availability and bioavailability in the soil for ideal plant development, impeding the plants from taking full advantage of their complementarity properties in the intercropping system. For instance, when nodulation occurred and peas were optimised for N-fixing, such as in the sand treatment, the intercropping system as it was designed in this experiment gave significant overall yield advantages, indicating the potential for such a method to be used to optimise crop growth on Mars.

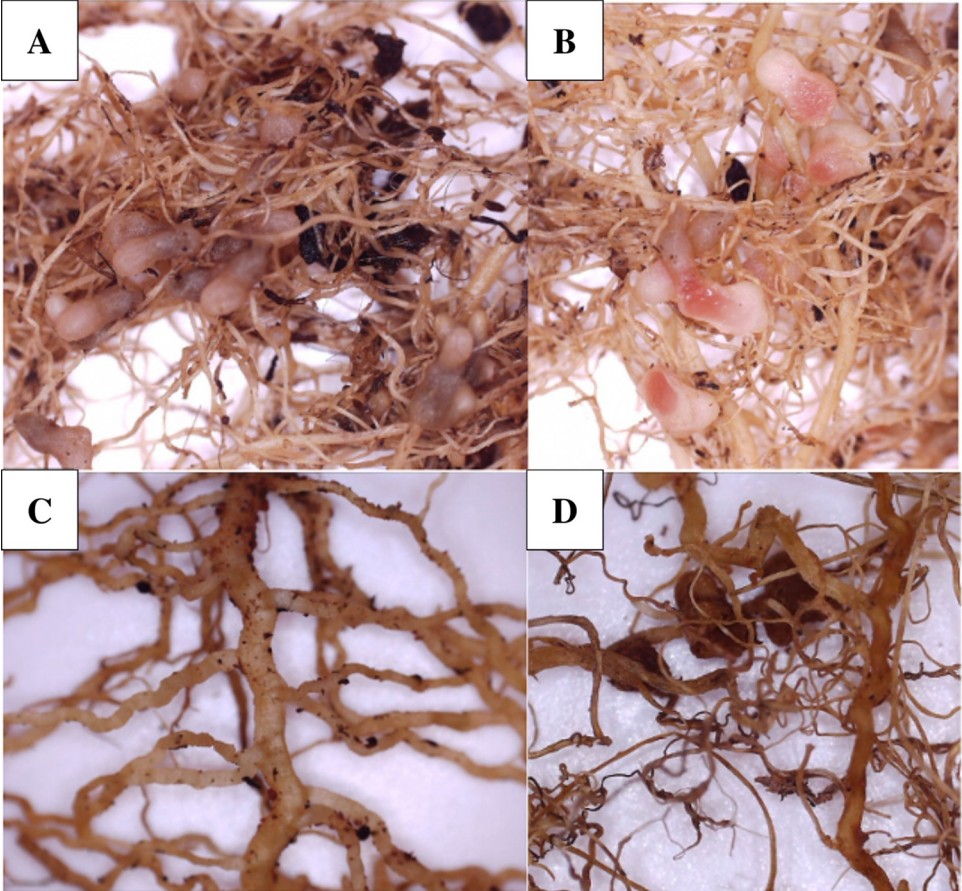

**Fig 8. Root and nodule comparison between soils.** A: Pea roots from Potting soil monocropping treatment, with intact nodules. B: Pea roots from sand monocropping treatment, with nodules dissected, where a pink coloration can be seen, indicating the nodules were active in N-fixing. C: Pea roots from Mars regolith simulant (intercropping treatment), showing thicker and more tortuous roots. D: Pea roots from Mars regolith (monocropping treatment), from the only replica that showed the presence of nodules. These roots also appear thicker and more tortuous.

Therefore, a chief solution for the improvement of the system would be to work on the sustainable amelioration of the Mars regolith, taking into account a realistic starting scenario on future Martian colonies. For example, initially selecting a higher grain grade of the regolith on Mars, at the onset of the Martian agricultural system, could potentially already aid in reducing soil compactness, improving drainage, salinity conditions, gas diffusion and nutrient availability.

Following the first crop harvest, using inedible parts to mix in as compost to the regolith could also greatly improve soil conditions. The increased organic matter would not only improve conditions for the survival and nodulation of Rhizobia bacteria, but also support the growth of all other species present by improving the regolith's nutrient content, bioavailability and uptake. Adding compost to MMS-1 Mars regolith results in higher plant biomass, with the highest results at 30:70 simulant:compost mixture [43]. Other combinations or varying ratios of candidate species can also be tested in order to find the optimum arrangement for the most efficient species complementarity. For example, a study by El-Gaid et al. [72] on intercropping tomato and bean found that the highest RYT values were achieved in a combination of 1 tomato plant: 3 bean plants (RYT = 1.26).

Of course, the absence of nodulation and reduced plant biomass and yield on the Mars regolith can be attributed not only to its physical properties and lack of micro-organisms and nutrients, but also to its intrinsic chemical composition. Further research would need to be conducted to isolate and identify which and how each of the chemical factors of MMS-1 would influence both in the nodulation of rhizobia as well as in the performance of plants.

Finally, actual Mars regolith and the initial conditions of a Martian colony would pose some additional challenges which, due to the emerging nature of this field of study, were purposefully "bypassed" here in order to reduce variables and produce more reliable results with respect to the focal study system. For example, we manually added a minimal amount of nutrients required for plant growth, disregarding the inherent sterility of the *in situ* regolith and presuming that such nutrients could be supplied from Earth or acquired through an established bioregenerative system on Mars. Furthermore, our simulant did not contain any perchlorates, a compound known to be present on the surface of Mars, nor did we analyse the effects of the regolith's heavy metals on plant growth. Although perchlorate and heavy metal presence can be remedied with the addition of plants or microbes to the soil [21,23], further research could include toxicity remediation and study the holistic effects that such measures would have on the microbial community and overall conditions for plant growth in the Martian regolith. As we build upon our collective knowledge for Martian agriculture, future studies can incorporate findings to produce a more overarching understanding of the complete scenario on Mars, including all its challenges and adopted solutions.

## Conclusions

In this study we sought to simulate an agricultural scenario likely to be encountered by early Martian colonists, such as the use of small pots, a controlled greenhouse environment, and a nutrient poor Mars regolith simulant with only essential nutrients added. Under such circumstances, we found that an intercropping system can be successful in optimising resource use efficiency if soil conditions are favourable to plant growth and nodulation of N-fixing bacteria, as we saw in the sand treatment (RYT = 1.32). On Mars regolith simulant, intercropping had an overall yield disadvantage compared to monocropping (RYT = 0.93). We postulated that this was most likely due to the absence of rhizobia nodulation in the Mars regolith, which negated the role of pea as a Nitrogen-fixer, impeding crops in the intercropping system from taking full advantage of their complementary properties. We identified that some of the physical and chemical properties of the Mars regolith simulant, such as elevated compactness and high pH, may have created a hostile environment for the survival and nodulation of rhizobia bacteria, while also limiting nutrient availability and bioavailability in the soil necessary for normal crop development. As further research, we suggest considering our proposed soil ameliorations to the Mars regolith, such as utilizing a higher grain grade to reduce soil compactness, and simulate a cyclic system in order to use past harvest waste as compost to increase soil pH and nutrient availability. We acknowledge that certain challenges with the actual regolith and starting conditions on Mars, such as the presence of perchlorates and the initial absolute sterility of the regolith, have been purposefully bypassed in order to maintain the focal objective of the study. Although solutions to such challenges exist, future studies should aim to ultimately integrate all these factors into a single system. As we build towards a more complete and comprehensive knowledge of Martian agriculture, focusing on improving regolith conditions could be key to unlocking the potential of rhizobia-legume interactions and species complementarity, and advance intercropping as a leading method to optimise fresh food production in future Martian colonies.

## Supporting information

**S1 Appendix. Comparison of the mineralogical, physical and chemical properties between the Mars regolith from the Rocknest eolian deposit in the Gale crater on Mars and the MMS-1 Mars regolith simulant.** Values are given in weight percentage (wt%). The table contains compiled data taken from multiple sources and own measurements.
(DOCX)

**S2 Appendix. pH, EC and nutrient contents of Hoagland nutrient solution.**
(DOCX)

## Acknowledgments

The authors wish to thank Zoë Berkers for help with measurements and harvest, and Paul Goedhart for his advice on the statistics.

## Author Contributions

**Conceptualization:** Rebeca Gonçalves, G. W. Wieger Wamelink, Jochem B. Evers.

**Data curation:** Rebeca Gonçalves, Peter van der Putten.

**Formal analysis:** Rebeca Gonçalves.

**Funding acquisition:** G. W. Wieger Wamelink, Jochem B. Evers.

**Investigation:** Rebeca Gonçalves.

**Methodology:** Rebeca Gonçalves, Peter van der Putten.

**Resources:** G. W. Wieger Wamelink, Jochem B. Evers.

**Supervision:** G. W. Wieger Wamelink, Peter van der Putten, Jochem B. Evers.

**Visualization:** Rebeca Gonçalves.

**Writing – original draft:** Rebeca Gonçalves.

**Writing – review & editing:** Rebeca Gonçalves, G. W. Wieger Wamelink, Jochem B. Evers.

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
