## [Decision Letter · Decision Letter 0]

13 Jul 2023

PONE-D-23-12961Intercropping on Mars: A Promising System to Optimize Fresh Food Production in Future Martian ColoniesPLOS ONE

Dear Dr. Gonçalves,

Thank you for submitting your manuscript to PLOS ONE. After careful consideration, we feel that it has merit but does not fully meet PLOS ONE’s publication criteria as it currently stands. Therefore, we invite you to submit a revised version of the manuscript that addresses the points raised during the review process. The manuscript is a good piece of research. However, before considering its possible acceptance, the details indicated in the comments must be corrected, and the questions raised by the reviewer must be explained.

We look forward to receiving your revised manuscript.

Kind regards,

Adalberto Benavides-Mendoza, Ph.D.

Academic Editor

PLOS ONE

Journal Requirements:

Additional Editor Comments:

According to the reviewer, the manuscript is a good piece of research. However, before considering its possible acceptance, the details indicated in the comments must be corrected, and the questions raised by the reviewer must be explained.

Reviewers' comments:

Reviewer's Responses to Questions

**Comments to the Author**

1. Is the manuscript technically sound, and do the data support the conclusions?

Reviewer #1: Yes

2. Has the statistical analysis been performed appropriately and rigorously? 

Reviewer #1: Yes

3. Have the authors made all data underlying the findings in their manuscript fully available?

Reviewer #1: Yes

4. Is the manuscript presented in an intelligible fashion and written in standard English?

Reviewer #1: Yes

5. Review Comments to the Author

Reviewer #1: Overall, the authors present a novel study with an interesting premise, good study design, and overall suitable metrics to measure their results. The growth of plants in-situ in regolith is a complex and relatively new topic and with limited studies on this subject, these studies will lay groundwork for later experiments. The use of agricultural metrics with intercropping in this paper bring a fresh approach to regolith studies and the use of a plant-rhizobium symbiosis builds on existing literature. The question and build up is mostly there to give context for intercropping in the introduction and the sources cited are relevant articles. However, one concern I have with this sort of research is that it does not address some of the crucial issues that growing things in-situ on Mars would encounter. That does not make this any less useful, but it maybe good to mention some of the other things that would need to be dealt with just to highlight where the focus of this research is. The data provided adequately supports the authors claims, but the discussion could use some elaboration to better understand how it fits the context of the field as a whole. The paper is an important contribution to a growing field and with a few minor revisions I would recommend this paper for publication. I have below outlined proposed revisions and additions to the article.

Major revisions:

None

Minor revisions:

Introduction

Generally speaking, it may be good to address some of the current literature on problems with simulant experiments that have been brought up. Issues like presence of perchlorates and other obstacles that maybe an issue should be briefly mentioned. On this note, addressing the soil properties of regolith a bit more in depth would help readers to understand why it is important to reestablish agricultural knowledge in this medium.

Line Revisions recommended

160-163 A justification for why Rhizobium Leguminosarum (RL) was used maybe helpful. This study does not address the specifics of the symbiosis nor study it in detail. There is nothing wrong with this as the focus of the study was elsewhere, but I would alter the language and explain the choice of using RL as a symbiote over something like Sinorhizobium meliloti which has been tested in multiple studies. Also, the species “meliloti” is spelled incorrectly in the manuscript.

Methods

Line Revisions recommended

257 Hoagland solution misspelled as Hoogland solution

Discussion

One thing to acknowledge in the discussion is the use of Hoagland solution and addition of soil to the regolith and sand. Native Martian regolith (and most simulants) have no nitrogen in them. Adding fertilizer like this inherently alters the fertility of the regolith and thus the focal study system. This maybe needed to support the targets of your study, but it needs to be acknowledged.

Line Revisions recommended

524-543 Finding some more sources about bacteria or bacterial symbiosis in regolith to help support this section would be excellent. It has a good start and I would say it is almost there but I think there are a few key aspects missing such as pH and presence of toxins in regolith.

Conclusions

Line Revisions recommended

596-598 I think this line needs some qualifying elements to it, as it is it overextends a bit. Under what conditions did this experiment occur? This is not regolith as such, there are issues to overcome. Something along the lines of “with the addition of X nutrients”.

Appendix

Line Revisions recommended

883 Table unreadable in PDF format

6. PLOS authors have the option to publish the peer review history of their article (what does this mean?). If published, this will include your full peer review and any attached files.

Reviewer #1: **Yes: **Franklin Harris

---

## [Author Response · Author response to Decision Letter 0]

24 Nov 2023

Dear Dr. Harris, 

Thank you for your kind feedback on our submitted manuscript. In our attached rebuttal letter, we have indicated in blue the actions taken to address each of your suggested revisions. 

We hope to have properly addressed your concerns.

Kind regards,

Rebeca Goncalves, on behalf of my Co-Authors

---

## [Decision Letter · Decision Letter 1]

25 Jan 2024

PONE-D-23-12961R1Intercropping on Mars: A Promising System to Optimise Fresh Food Production in Future Martian ColoniesPLOS ONE

Dear Dr. Gonçalves,

Thank you for submitting your manuscript to PLOS ONE. After careful consideration, we feel that it has merit but does not fully meet PLOS ONE’s publication criteria as it currently stands. Therefore, we invite you to submit a revised version of the manuscript that addresses the points raised during the review process.

 The manuscript still requires a specific abstract, introduction, and discussion improvements. Please take into account the comments and recommendations of Reviewer 2.

We look forward to receiving your revised manuscript.

Kind regards,

Adalberto Benavides-Mendoza, Ph.D.

Academic Editor

PLOS ONE

Journal Requirements:

Additional Editor Comments:

The manuscript still requires a specific abstract, introduction, and discussion improvements. Please take into account the comments and recommendations of Reviewer 2.

Reviewers' comments:

Reviewer's Responses to Questions

**Comments to the Author**

1. If the authors have adequately addressed your comments raised in a previous round of review and you feel that this manuscript is now acceptable for publication, you may indicate that here to bypass the “Comments to the Author” section, enter your conflict of interest statement in the “Confidential to Editor” section, and submit your "Accept" recommendation.

Reviewer #1: All comments have been addressed

Reviewer #2: All comments have been addressed

2. Is the manuscript technically sound, and do the data support the conclusions?

Reviewer #1: Yes

Reviewer #2: Yes

3. Has the statistical analysis been performed appropriately and rigorously? 

Reviewer #1: Yes

Reviewer #2: Yes

4. Have the authors made all data underlying the findings in their manuscript fully available?

Reviewer #1: Yes

Reviewer #2: Yes

5. Is the manuscript presented in an intelligible fashion and written in standard English?

Reviewer #1: Yes

Reviewer #2: Yes

6. Review Comments to the Author

Reviewer #1: I see a lot of improvement from the last time I read this manuscript. I think that in terms of the science, you have adequately discussed your methods, and brought a better focus on the rhizobium-plant symbiosis in the methods. The discussion also better highlights the limitations of regolith which will help your results fit better into existing literature. While this is still a new field, I have found that the literature selected and discussed to explain some of the observed differences was quite representative of the state of the field. Additionally, the conclusion also highlights limitations as previously noted.

Good job and best wishes on your endeavors.

Reviewer #2: Reviewer:

- Comments to the Author:

- The article is interesting and the results look original, but serious deficiencies have been observed:

- Title is acceptable.

- Abstract:

- 1) Underscore the scientific value-added to your paper in your abstract. Your abstract should clearly state the essence of the problem you are addressing, what you did and what you found and recommend. That will help a prospective reader of the abstract to decide if they wish to read the entire article.

- 2) Please add the values of the traits mentioned in the abstract section.

-Keywords is acceptable

Introduction:

- The introduction need more focus on mechanisms instead of narration.

- Clearly state whether your objective is novel or no.

- What is your research innovation and novelty?

- The introduction need more focus on mechanisms instead of narration.

- Please write the research hypotheses at the end of the introduction.

Material and method:

- In the materials and methods section, please remove irrelevant and redundant descriptions. Some of the mentioned explanations should be moved to the introduction section. For example, delete lines 245 to 249.

- Add references to the material and method of the traits being measured.

Results and discussion:

- Do not re-report the results in the discussion section.

- Conclusion is acceptable.

References

- The number of references is too high, please remove some references.

Kind regards,

Reviewer

7. PLOS authors have the option to publish the peer review history of their article (what does this mean?). If published, this will include your full peer review and any attached files.

Reviewer #1: **Yes: **Franklin L. Harris

Reviewer #2: **Yes: **Esmaeil Rezaei-Chiyaneh

---

## [Author Response · Author response to Decision Letter 1]

18 Mar 2024

Dear Dr. Rezaei-Chiyaneh, 

Thank you very much for your kind feedback on our submitted manuscript. 

We have indicated in the following document the actions taken to address each of your suggested revisions. 

These are right next to your highlighted points under “item 6 – Review Comments to the Author”.

Please note that the line numbers refer to the document with track changes.

We hope to have properly addressed your concerns.

Kind regards,

Rebeca Gonçalves

---

## [Editor Report · Decision Letter 2]

28 Mar 2024

Intercropping on Mars: A Promising System to Optimise Fresh Food Production in Future Martian Colonies

PONE-D-23-12961R2

Dear Dr. Gonçalves,

We’re pleased to inform you that your manuscript has been judged scientifically suitable for publication and will be formally accepted for publication once it meets all outstanding technical requirements.

Kind regards,

Adalberto Benavides-Mendoza, Ph.D.

Academic Editor

PLOS ONE

Additional Editor Comments (optional):

The authors resolved all concerns and suggestions; therefore, the manuscript can be accepted.
---

## [Editor Report · Acceptance letter]

5 Apr 2024

PONE-D-23-12961R2 

PLOS ONE

Dear Dr. Gonçalves, 

I'm pleased to inform you that your manuscript has been deemed suitable for publication in PLOS ONE. Congratulations! Your manuscript is now being handed over to our production team.

Kind regards, 

on behalf of

Dr. Adalberto Benavides-Mendoza 

Academic Editor

PLOS ONE